# Case Study of Transient Dynamics in a Bypass Reach

**Anton J. Burman [1],\*** , **Anders G. Andersson [1]** , **J. Gunnar I. Hellström [1]** and **Kristian Angele [2]**

1   Division of Fluid and Experimental Mechanics, Luleå Tekniska Universitet, 971 87 Luleå, Sweden;
    anders.g.andersson@ltu.se (A.G.A.); gunnar.hellstrom@ltu.se (J.G.I.H.)
2   Vattenfall Research and Development, Älvkarlebylaboratoriet, 814 70 Älvkarleby, Sweden;
    kristian.angele@vattenfall.com
\*   Correspondence: anton.burman@ltu.se

**Abstract:**  The operating conditions of Nordic hydropower plants are expected to change in the coming years to work more in conjunction with intermittent power production, causing more frequent hydropeaking events. Hydropeaking has been shown to be detrimental to wildlife in the river reaches downstream of hydropower plants. In this work, we investigate how different possible future hydropeaking scenarios affect the water surface elevation dynamics in a bypass reach in the Ume River in northern Sweden. The river dynamics has been modeled using the open-source solver Delft3D. The numerical model was validated and calibrated with water-surface-elevation measurements. A hysteresis effect on the water surface elevation, varying with the downstream distance from the spillways, was seen in both the simulated and the measured data. Increasing the hydropeaking rate is shown to dampen the variation in water surface elevation and wetted area in the most downstream parts of the reach, which could have positive effects on habitat and bed stability compared to slower rates in that region.

**Keywords:** inherent damping; hydropeaking; river dynamics; hydraulic modeling; delft3d

## 1. Introduction

When the Paris Agreement was signed in 2016, most of the world committed to reducing carbon dioxide emissions in order to keep global warming temperatures below two degrees Celsius compared to preindustrial levels [1]. In response, the governments of the Nordic countries have declared different emission goals in the coming decades. The Swedish government has pledged to have net zero greenhouse gas emissions by 2045 [2]. Similarly, the greenhouse gas emissions are to be reduced by 50% in Norway by 2030 [3]. In Finland, the goal is to cut emissions with 39% by 2030 in comparison with the emissions in 2005 [4]. On a larger scale, the European Council aims to cut at least 40% of the greenhouse gas emissions compared to 1990 as well as have 32% renewable energy [5]. The share of renewable energy production increased from 9.6% to 18.9% between the years 2004 and 2018 and is expected to increase more in the coming years [6]. Most of the renewable energy produced in Europe is either hydropower (mainly the Nordic countries) or intermittent power sources such as wind power and solar power [6]. Currently, the further integration of the Norwegian and Swedish power grids with mainland Europe is being planned [7]. One of the grid integration projects that is already on going is The North Sea Link, connecting the Norwegian and British power grids, which is expected to be finished in 2021 [8]. The nature of hydropower makes it convenient to store energy in times of favorable conditions for intermittent power production. When the weather changes and the conditions become less favorable, hydropower can be used as a complement to stabilize the power grid. In order for this to be achievable on a European scale, the role of Nordic hydropower is expected to change to be more aligned with the power production needs of mainland Europe rather than producing power

mainly for consumption in the Nordic countries. This in turn will affect the operating conditions in Nordic hydropower plants, causing more hydropeaking events and rapidly fluctuating water levels. It was with this background that the HydroFlex consortium was established with an overarching goal of researching scenarios with as many as 30 starts and stops per day [9]. It is well established that hydropeaking can be detrimental to downstream river reaches. Diurnal flow patterns downstream of hydropower plants increase the stranding of macroinvertebrates as well as reducing the species richness of benthic macroinvertebrates [10]. The negative impacts of hydropeaking on different fish species have been investigated all across the globe. Studies in Norway—both in a laboratory environment as well as in a river—have been performed, investigating the factors causing stranding for Atlantic salmon (*Salmo salar*) and brown trout (*Salmo trutta*) during rapid dewatering [11,12]. Temperature, season, and lighting conditions were found to impact the stranding rate [11]. The stranding of juvenile brown trout was minimized when the rate of water level change was reduced from >60 cm/h to <10 cm/h [12]. In the USA, a study showed that a more stable flow regime led to greater abundance of rainbow trout (*Oncorhynchus mykiss*) [13]. Hydropeaking was shown to decrease the Composite Suitability Index and the Weighted Suitable Area for pale chub (*Zacco platypus*) in South Korea [14]. Hydropeaking also affects the river margin erosion as well as the river morphology [15]. Hydropeaking could also negatively affect human safety [16]. There are ways of reducing the impact of hydropeaking. The most obvious is to change the operating conditions to reduce the number of hydropeaking events [17]. Another approach is to modify the structure around the tailrace. One approach that has been suggested is to divert some of the discharge during hydropeaking before the tailrace and to gradually introduce it to the main river downstream [18]. It has also been suggested that discharge water can be temporarily stored in an Air Cushion Underground Reservoir (ACUR) and gradually released into the tailrace in times of no hydropeaking [9,19]. An additional approach could be to use the inherent inertia in the river to reduce the impact of hydropeaking in some stretches of a river. The delay due to inherent inertia in water-surface elevation as a function of distance from the tailrace has been documented [20,21]. In this study, the open-source hydrodynamics solver Delft3D is used for modeling the flow in the river. Delft3D has been used for a wide variety of hydrodynamic problems such as morphodynamics in a tidal river [22], braided river flows [23], and tidal dynamics in a mangrove creek catchment [24]. The aim of the work presented here is to investigate different hydropeaking-frequency scenarios in a bypass reach in the Ume River in northern Sweden as well as studying the transient dynamics in the reach including the hysteresis for the water-surface elevation (WSE) and the wetted area.

## 2. Theory

### 2.1. Governing Physics

The governing equations of all fluid dynamics are the Navier–Stokes equations. The incompressible Navier–Stokes equations consists of three momentum equations

$$\frac{\partial \mathbf{u}}{\partial t} + (\mathbf{u} \cdot \nabla)\mathbf{u} = -\frac{\nabla p}{\rho} + \mathbf{F} + \nu\nabla^2\mathbf{u}, \tag{1}$$

and one continuity equation

$$\nabla \cdot \mathbf{u} = 0, \tag{2}$$

where $\mathbf{u}$ is the velocity vector, $p$ is the total pressure, $\rho$ is the density of the fluid, $\mathbf{F}$ is the sum of body forces on the system, and $\nu$ is the kinematic viscosity [25]. The Navier–Stokes equations are a nonlinear set of partial differential equations that pose difficulties when solved numerically. The nature of turbulence is such that all length scales in the flow needs to be resolved. This is problematic when, as in rivers, the length scales can be on the order of magnitude of tenths of kilometers. Most commonly in computational fluid dynamics (CFD) the turbulence is modeled using Reynolds averaging. Many of the most commonly used turbulence models, such as k-ε, SST, and RSM are based on this method. Another approach is to model the smaller length scales using subgrid models and resolving the larger

scales, as is done in LES approaches [26]. Both these methods are often too computationally demanding for large scale river simulations. One way to simplify the Navier–Stokes equations is by deriving the two-dimensional Shallow-Water Equations (SWEs) by assuming that the pressure is almost hydrostatic and that the horizontal length scales are significantly larger than the depth length scales [27]. The SWEs contain two momentum equations and one continuity equation

$$\frac{\partial u}{\partial t} + u\frac{\partial u}{\partial x} + v\frac{\partial u}{\partial y} = -g\frac{\partial \zeta}{\partial x} + F_x, \tag{3}$$

$$\frac{\partial v}{\partial t} + u\frac{\partial v}{\partial x} + v\frac{\partial v}{\partial y} = -g\frac{\partial \zeta}{\partial y} + F_y, \tag{4}$$

$$\frac{\partial \zeta}{\partial t} + \frac{\partial (hu)}{\partial x} + \frac{\partial (hv)}{\partial y} = 0, \tag{5}$$

where $F_x$ and $F_y$ are the $x$ and $y$ components of the body forces on the system, $g$ is the gravitational acceleration, $h$ is the depth, and $\zeta$ is the displacement of the water surface.

*2.2. Implementations in Delft3D*

2.2.1. Physics

In Delft3D, the SWEs are formulated in orthogonal curvilinear coordinates. The continuity equation is

$$\frac{\partial \zeta}{\partial t} + \frac{1}{\sqrt{G_\xi}\sqrt{G_\eta}}\frac{\partial \left( (d+\zeta)U_\xi \right)}{\partial \xi} + \frac{1}{\sqrt{G_\xi}\sqrt{G_\eta}}\frac{\partial \left( (d+\zeta)U_\eta \right)}{\partial \eta} = (d+\zeta)Q, \tag{6}$$

where $G_\xi$ and $G_\eta$ are transformation coefficients between curvilinear and orthogonal coordinates, $d$ is the depth, $U_\xi$ and $U_\eta$ are the depth-averaged velocities in the respective direction, and $Q$ is the contribution per unit area due to the discharge or withdrawal of water, precipitation, and evaporation. The momentum equations in $\xi$ and $\eta$ directions are then

$$\frac{\partial U_\xi}{\partial t} + \frac{U_\xi}{\sqrt{G_\xi}}\frac{\partial U_\xi}{\partial \xi} + \frac{U_\eta}{\sqrt{G_\eta}}\frac{\partial U_\xi}{\partial \eta} - \frac{U_\eta^2}{\sqrt{G_\xi}\sqrt{G_\eta}}\frac{\partial \sqrt{G_\eta}}{\partial \xi} + \frac{U_\xi U_\eta}{\sqrt{G_\xi}\sqrt{G_\eta}}\frac{\partial \sqrt{G_\xi}}{\partial \eta} - fU_\eta = -\frac{P_\xi}{\rho\sqrt{G_\xi}} + F_\eta + M_\eta \tag{7}$$

and

$$\frac{\partial U_\eta}{\partial t} + \frac{U_\xi}{\sqrt{G_\xi}}\frac{\partial U_\eta}{\partial \xi} + \frac{U_\eta}{\sqrt{G_\eta}}\frac{\partial U_\eta}{\partial \eta} - \frac{U_\xi^2}{\sqrt{G_\xi}\sqrt{G_\eta}}\frac{\partial \sqrt{G_\xi}}{\partial \eta} + \frac{U_\xi U_\eta}{\sqrt{G_\xi}\sqrt{G_\eta}}\frac{\partial \sqrt{G_\eta}}{\partial \xi} + fU_\xi = -\frac{P_\eta}{\rho\sqrt{G_\eta}} + F_\xi + M_\xi, \tag{8}$$

where $P_\eta$ and $P_\xi$ are pressure gradients, $F_\xi$ and $F_\eta$ are unbalanced horizontal Reynolds stresses, $M_\xi$ and $M_\eta$ are contributions due to external sources of momentum, and $f$ is the Coriolis parameter. [28]

2.2.2. Numerics

Delft3D uses finite differences as its method of discretization. Furthermore, the mesh is staggered. Staggered grids have advantages when solving the SWEs—such as, the boundary conditions are easier to implement and that staggered grids have also been shown to reduce oscillations in the water level. The solver is using an alternating-direction-implicit (ADI) method for the time integration. In each time step, the nonlinear terms in the momentum equations are linearized and solved iteratively in order to ensure continuity in each timestep. All discretizations are at least second-order-accurate [28].

*2.3. Richardson Extrapolation*

In 1911, Richardson suggested an approach to quantify the numerical errors that arise from discretizing partial differential equations [29]. In his seminal paper, he specifically applied it to stresses

on a masonry dam, but the method can be used for any numerical approach. By evaluating some variable on several meshes of varying size one can find an approximate grid-independent value [26,30]. The first step is to define a representative grid size for at least three different meshes. There are many ways to define the representative grid size, one possible definition for two-dimensional grids is

$$h = \left[ \frac{1}{N_x N_y} \right]^{1/2},$$

(9)

where $N_x$ is the number of nodes in the $X$ direction and $N_y$ is the number of nodes in the $Y$ direction. The next step is to perform simulations on the chosen grids and extract a representative variable of choice $\phi$. Then, the variables $r_{32} = h_3/h_2$, $r_{21} = h_2/h_1$, $\varepsilon_{32} = \phi_3 - \phi_2$ and $\varepsilon_{21} = \phi_2 - \phi_1$ are defined. Now, the apparent order of the solution can be computed with the implicit equation

$$\frac{1}{r_{21}} \left| ln \left| \frac{\varepsilon_{32}}{\varepsilon_{21}} \right| + ln \left( \frac{r_{21}^p - s}{r_{32}^p - s} \right) \right| - p = 0,$$

(10)

where $p$ is the apparent order and

$$s = sign \left( \frac{\varepsilon_{32}}{\varepsilon_{21}} \right).$$

(11)

Then, the extrapolated grid-independent value can be expressed as

$$\phi_{ext} = \frac{r_{21}^p \phi_1 - \phi_2}{r_{21}^p - 1}.$$

(12)

## 3. Materials and Methods

### 3.1. Study Site

The chosen study site is the bypass reach at the Stornorrfors hydropower plant in the Ume River. Stornorrfors is the hydropower plant that, on average, produces the most electricity annually in Sweden [31]. During summer, there is a minimum flow in the reach of 21 m$^3$/s for upstream fish migration, during weekends the flow in the reach is increased to 50 m$^3$/s for aesthetic reasons. In the winter, the reach is mostly dry. The spillways and the fishway spill into the most upstream part of the study reach, the bypass joins the tailrace of the power plant shortly downstream of the study reach, see Figure 1. The reach between the spillways and the confluence is approximately 7-km-long. The entirety of the Ume River is regulated, while the Vindel River—a tributary to the Ume River that merges a couple of kilometers upstream of the study reach—is not regulated. During the spring flood, it is therefore common that spilling occurs in the bypass reach. It would be inaccurate to describe the flow conditions in the study reach as hydropeaked since the discharge in the reach is not necessarily related to the power production of the power plant. The study reach is however subject to rapid changes in discharge, partly during the spring flood and partly during the weekly increase and decrease in discharge.

### 3.2. Bathymetry and Depth Measurements

The bathymetry of the study reach was measured during winter when most of the reach is dry. This makes it possible to measure the bathymetry with drone stereo photogrammetry. This method has spatial accuracy on the order of a decimeter. Along the reach, eight pressure loggers (divers) were positioned, their position in the reach can be seen in Figure 1. The water-level measurements were performed from mid-May to mid-July in the summer of 2017. The placement of the divers in the reach was partly decided based on accessibility to the reach and partly by variation of cross-section shape, hence the sparser placement in the central parts of the reach (points 5–7). The GPS coordinates of the divers can be seen in [20]. The temporal resolution of the divers is one minute. The uncertainty of the

water-level measurements is 2 cm [32]. The absolute positioning for both the photogrammetry and the divers was measured with a GPS pole [32]. The data collected from the drone measurements were then processed in ArcGIS to create a Digital Elevation Model (DEM), the model is reported in Figure 2a. In Figure 2b a typical increase–decrease cycle for each validation point is plotted. Only the periods of increase and decrease have been taken into consideration, hence, the long periods of intermediate steady states were discarded. There are some abnormalities in the DEM due to ice build-up at the time of measurements, these can be seen as deep holes in the DEM in Figure 2a. The deepest hole reaches $-8$ [*MASL*] and occurs close to validation point 7.

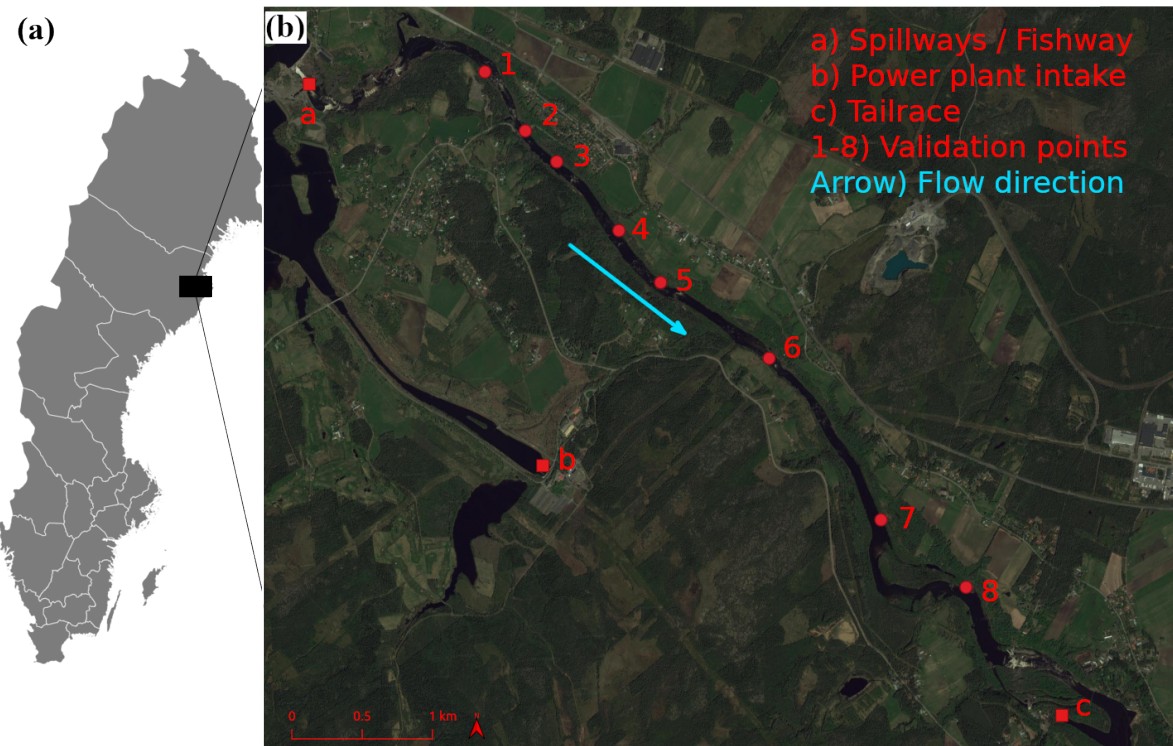

**Figure 1.** (**a**) Position of Stornorrfors in Sweden. (**b**) Key locations in the study reach and the extent of the numerical model.

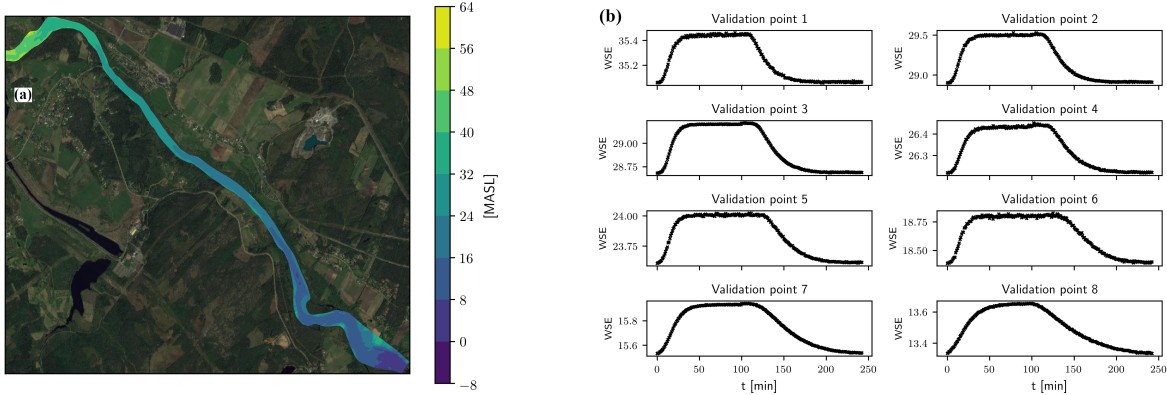

**Figure 2.** Field measurements in the study reach: (**a**) Digital elevation model of the study reach in meters above sea level (MASL). (**b**) Water-surface elevation (WSE) in all eight validation points during a typical increase–decrease scenario.

### 3.3. Scenarios

#### 3.3.1. Hysteresis Scenarios

Hysteresis is an effect that occurs for some nonlinear systems. In systems where hysteresis occurs, the function value is dependent on the history of the system; in this case, it is whether the water level is increasing or decreasing [33]. Hysteresis occurs for many hydrological variables such as river discharge, solute concentration, and suspended sediment concentration [34]. Hysteresis also occurs in discharge rating curves in rivers [35]. Two simulations were performed to investigate the hysteresis effect on the WSE. The WSE was considered in two legs, one where the WSE is decreasing and one where it is increasing. In the first case, the discharge was reduced from 50 m$^3$/s to 21 m$^3$/s in 5 min. In the second case, the discharge was increased from 21 m$^3$/s to 50 m$^3$/s in 5 min. In both simulations, a steady state was ensured both before and after the change in discharge.

#### 3.3.2. Hydropeaking Scenarios

Six hydropeaking scenarios with different inlet discharge conditions were considered, see Figure 3. The scenarios each span six hours and consider different possible start and stop schemes. The scenarios correspond to a change in discharge 10, 20, 30, 40, 50, and 60 times per day, and the flow changes were evenly distributed throughout the day. The corresponding opening and closing times were five minutes in all scenarios, since this was observed to be the most common in the public discharge data [36]. In the calibration and the mesh study, the scenario is a constant discharge of 50 m$^3$/s.

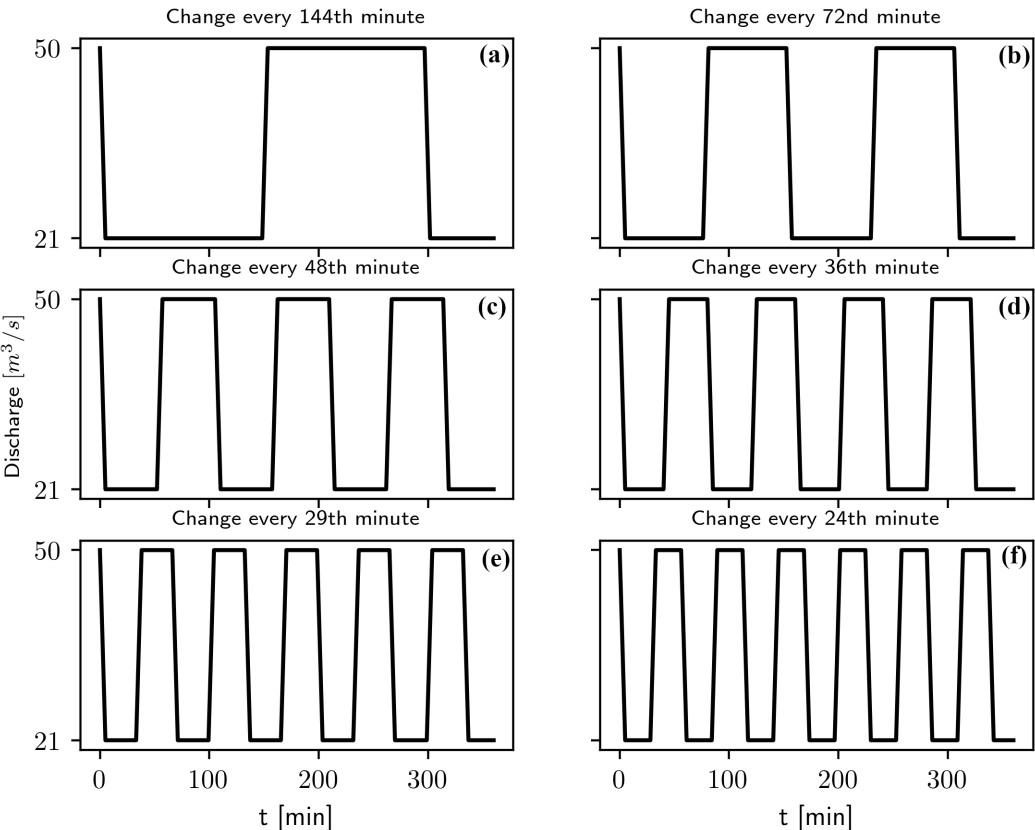

**Figure 3.** Hydrograph for the six different scenarios under consideration: (**a**) 10 flow changes per day. (**b**) 20 flow changes per day. (**c**) 30 flow changes per day. (**d**) 40 flow changes per day. (**e**) 50 flow changes per day. (**f**) 60 flow changes per day.

### 3.4. Calibration

The roughness of the model was calibrated by performing a parametric sweep of the Manning number for a steady-state discharge of 50 m³/s. The Manning number was swept from 0.03 s/m$^{1/3}$ to 0.1 s/m$^{1/3}$, which corresponds to the extreme values of the Manning number in natural channels [37]. In each of the simulations, the Manning number was kept constant in the entire reach. This approach has been used with success in other studies [38]. It is assumed that the reach in proximity of the validation points are of the same roughness to the validation point. By comparing the simulated water levels to the measured diver data, it was then possible to find the Manning number that produced the smallest error [20]. The calibrated WSE is plotted against the measured WSE in Figure 4. Relevant statistics can be seen in Table 1. The Pearson correlation of 0.9995 obtained in this study is comparable to the ones obtained in [21,38]. The corresponding Manning number distribution can be seen in Figure 4.

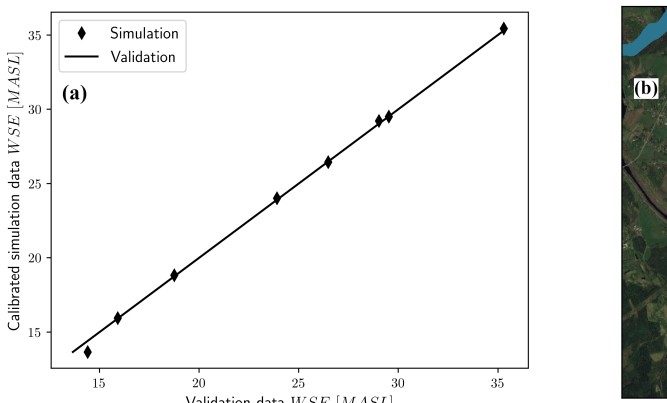
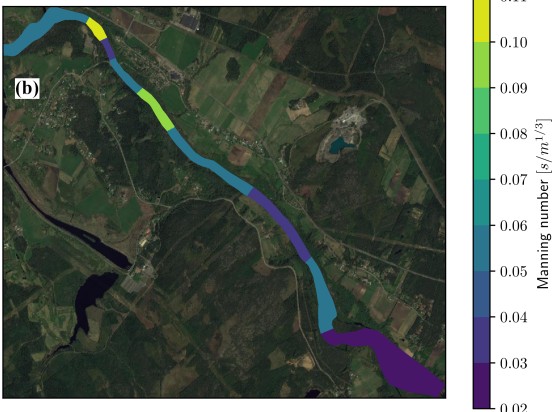

**Figure 4.** Outcome of calibration in the study reach. (**a**) Correlation plot between validation data and simulated WSE. (**b**) Manning number distribution obtained from calibration.

**Table 1.** Statistics regarding the calibration of the model.

| Property | Value |
|---|---|
| Maximum error | 0.74 [m] |
| Minimum error | 0.02 [m] |
| Median error | 0.08 [m] |
| Standard deviation | 0.304 [m] |
| Pearson correlation | 0.9995 |

### 3.5. Model Setup

Three boundary conditions are required, one for the upstream inlet of the reach, one for the downstream outlet, and one for the slip behavior of the wall. The upstream condition was set to "total discharge", where the hydrographs in Figure 3 were used. For the mesh study, the total discharge was set to 50 m³/s. At the downstream boundary, the condition was set to "Neumann" with a value of 0.001 for the water surface. The slip condition was set to "free slip", which for large scale hydrodynamic simulations, is a reasonable assumption [28]. Both boundaries had a reflection parameter of 0. The bathymetry seen in Figure 2 was interpolated on all the meshes using the QUICKIN interpolation tool [39]. The threshold depth was set to 0.1 m and the advection scheme used was "cyclic", which is the standard advection scheme in Delft3D. The Manning roughness formula was chosen with a roughness file, the distribution can be seen in Figure 4. A timestep of $t = 0.005$ min proved to be sufficient to obtain stable solutions.

### 3.6. Wetted Area Calculation

The wetted area is not a variable that can be exported natively from Delft3D. Instead, a method to compute the wetted area was implemented. Each time the area was calculated, the WSE was exported as a 400-DPI image. The 400-DPI images proved to be sufficiently resolved. Afterwards, the number of pixels that contained any data was counted. For each mesh, the extent of the $X$ and $Y$ coordinates were extracted. With these variables, it was possible to compute the wetted area as

$$A_{wetted} = \frac{\text{Nr. of active pixels}}{\text{Nr. of total pixels}} A_{total}, \tag{13}$$

where $A_{total}$ is the total area spanned by the square defined as

$$A_{total} = (X_{max} - X_{min})(Y_{max} - Y_{min}), \tag{14}$$

where $X_{max}$ and $X_{min}$ is the maximum and minimum extent of the $X$ coordinate respectively, and $Y_{max}$ and $Y_{min}$ are the respective maximum and minimum extents of the $Y$ coordinate. The black borders in Figure 5 correspond to $A_{total}$.

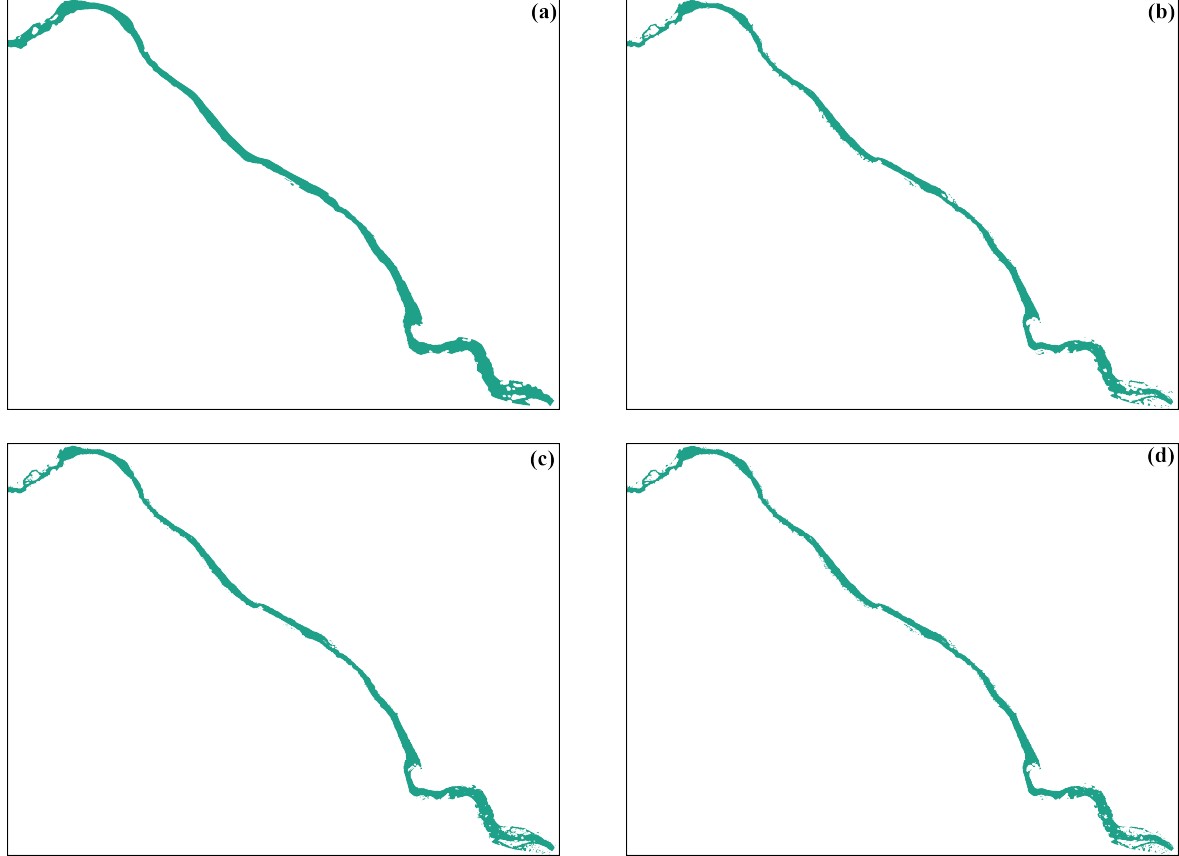

**Figure 5.** The 400-DPI image files generated for $A_{wetted}$ calculation in the mesh study. The black borders correspond to $A_{total}$. (**a**) Coarse mesh. (**b**) Less-fine mesh. (**c**) Finer mesh. (**d**) Finest mesh.

### 3.7. Mesh Study

Four different meshes of varying sizes were used for the mesh study. The meshes are denoted as *coarse*, *less-fine*, *finer*, *finest*. A table of the mesh properties can be seen in Table 2. By studying Figure 5, it is apparent that the coarse mesh does not properly capture the complexity of the bathymetry. This is especially noticeable in the most upstream and downstream parts of the reach where there are significant rapids. This in turn means that coarser grids will overestimate the wetted area.

As the grid is refined, it is noticed that more of the bathymetry is captured and the wetted area decreases. The resulting wetted areas are tabulated in Table 3 along with the results of the Richardson extrapolation. The Richardson extrapolated value was obtained using the three finest meshes and solving Equation (12) using the Scientific Python optimization module [40].

**Table 2.** Mesh properties for the four different meshes used in the study. Representative size is defined in Equation (9).

| Grid | $N_x$ | $N_y$ | Nr. of Elements | Representative Size [1/m] |
|------|-------|-------|-----------------|---------------------------|
| Coarse | 521 | 26 | 13546 | 0.0086 |
| Less-Fine | 1040 | 74 | 76960 | 0.0036 |
| Finer | 2078 | 218 | 453004 | 0.0015 |
| Finest | 4154 | 218 | 905572 | 0.0011 |

**Table 3.** Wetted area for all meshes and Richardson extrapolated area. Error is given in percentage of the Richardson extrapolated value.

| Grid | $A_{wetted}$ [m$^2$] | Error |
|------|----------------------|-------|
| Coarse | 863897 | +26.10% |
| Less-Fine | 724442 | +5.71% |
| Finer | 746164 | +8.88% |
| Finest | 733532 | +7.03% |
| Richardson Extrapolation | 685334 | - |
| Standard Deviation | 0.304 | |

The mesh with the smallest error turned out to be the *less-fine* mesh. However, there is uncertainty in how well this mesh resolves the bathymetry. By studying the area for the *finer* and *finest* meshes, it is apparent that the features are similar—both the shape of the contour as well as the size of $A_{wetted}$. This is not necessarily true for the *less-fine* mesh. For this reason, the "finer" grid was chosen, since this grid provided similar accuracy to the *finest* grid.

## 4. Results And Discussion

### 4.1. WSE Hysteresis

The WSE hysteresis response is plotted in Figure 6. Measures of time for the simulations and the measurements for each leg have been tabulated in Table 4. In order for each loop to be comparable in each validation point, the WSE was normalized according to

$$WSE_{norm} = \frac{WSE - WSE_{t=\infty}}{WSE_{t=0} - WSE_{t=\infty}}, \tag{15}$$

for the leg where the WSE is decreasing and

$$WSE_{norm} = \frac{WSE - WSE_{t=0}}{WSE_{t=\infty} - WSE_{t=0}}, \tag{16}$$

for the leg where the WSE is increasing. $WSE_{t=0}$ is the steady state WSE at $t = 0$, and $WSE_{t=\infty}$ is the steady state WSE once the new steady state is obtained in each respective validation point. The new steady state was considered as reached when the $WSE$ reached 99% of $WSE_{t=0}$ and $WSE_{t=\infty}$, respectively.

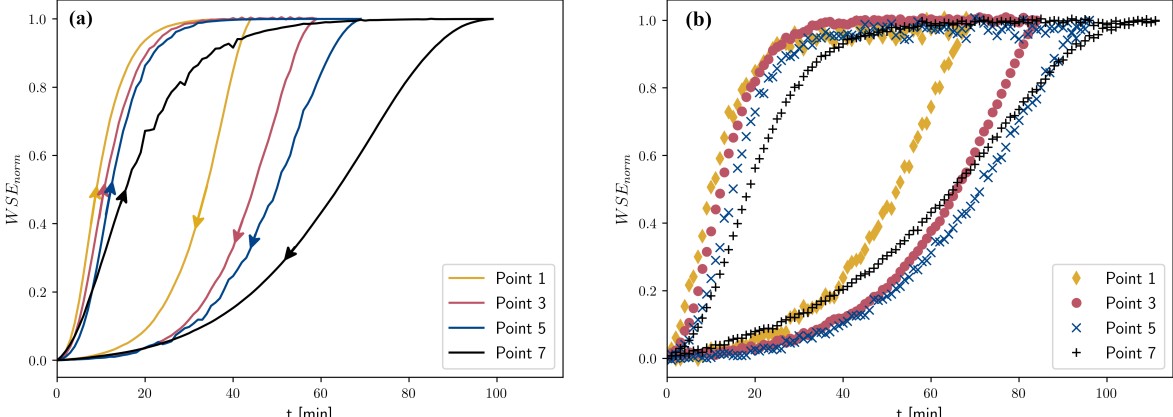

**Figure 6.** Hysteresis loop for $WSE_{norm}$ in validation points 1, 3, 5, and 7, given the measured scenarios in Figure 2 and the simulated scenarios described in Section 3.3.1. (**a**) Simulated hysteresis loop. The arrows indicate the direction of the process. (**b**) Measured hysteresis loop.

**Table 4.** Time for each leg of the hysteresis loop for $WSE_{norm}$ obtained from Figure 6. The time it takes for the $WSE$ to reach the new steady state is referred to as *increase time* and *decrease time*. All units of time are in minutes.

| Validation Point | Simulated | | Measured | |
|---|---|---|---|---|
| | WSE Increase Time | WSE Decrease Time | WSE Increase Time | WSE Decrease Time |
| Point 1 | 29 | 44 | 32 | 69 |
| Point 3 | 33 | 56 | 35 | 79 |
| Point 5 | 35 | 69 | 36 | 88 |
| Point 7 | 62 | 99 | 51 | 109 |

The general shape of the hysteresis loops are similar for both the simulated and the measured cases. It is apparent that there is a dampening dispersion effect on both the *increase time* leg and the *decrease time* leg, this is seen in both the measured and the simulated responses in Figure 6. The manifestation of the dampening is an increase in time for each leg and the effect is magnified with downstream coordinate, as can be seen in Table 4. The simulated *increase time* predicts the measured time accurately for points 1, 3, and 5. The discrepancy at point 7 could be due to poor geometry data in proximity to that point, see Figures 1 and 2. The *decrease time* leg appears to have a systematic delay for each point. No change in the hysteresis was seen when simulations with horizontal LES or "no-slip" conditions were used. There are a couple of possible explanations for this difference. At least seven tributaries of varying size connect to the reach, the inflow of these tributaries were not taken into consideration when modeling. The weather conditions during the measurements were not taken into consideration, hence, evaporation effects and precipitation were not taken into account. Furthermore, there is uncertainty in how fast the closing and opening time of the spillway gate is. Another explanation could be that the *decrease time* leg is fundamentally three-dimensional in nature and is therefore not properly represented by two-dimensional models. Vertical mixing effects caused by scattered rocks and other stochastic elements in the reach could be important to resolve to get the full picture and capture all dynamic effects of this leg [41]. In contrast, the *increase time* leg is dominated by the initial positive surge caused by the upstream increase in discharge, whose behavior is more readily captured using SWE.

*4.2. WSE Dynamics with Different Scenarios*

The WSE in points 1, 2, 3, and 4, given the hydrograph scenarios in Figure 3, have been plotted in Figures 7 and 8. Similarly, the WSE for the points 5, 6, 7, and 8 can be seen in Figures 9 and 10. It is apparent that the number of flow changes per day greatly impacts the transient dynamics in the reach. For the case with 10 flow changes per day, it is observed that the WSE reaches the steady state

corresponding to $Q$ = 50 m$^3$/s and 21 m$^3$/s in all validation points, see Figures 7 and 9. Similarly for the 20 flow changes per day case, we see that the respective steady state is reached in all validation points except point 8. In this point, there is a state of continuous dynamical change where the WSE never reaches any resemblance of steady state. This effect is noticed for all scenarios except the case of 10 changes per day. For the case of 20 flow changes per day this point of continuous dynamics occurs somewhere between point 7 and point 8. Analogously, this point is between point 5 and point 6 for the 30 flow changes case, see Figure 9. For the case of 40 flow changes per day, the point occurs between point 3 and point 4, see Figure 8. Further, for the 50 and 60 flow changes per day cases, it occurs somewhere upstream of point 1. For the points 5–8 for the 40, 50, and 60 flow changes per day cases (Figure 10), the hysteresis behavior seen in Figures 7 and 9 is no longer observed, rather, the WSE appears to oscillate sinusoidally. One explanation for this behavior could be that the time scales become comparable or smaller than the *decrease time* and *increase time* legs seen in Table 4. It is also noticed that in some cases the steady state for the increasing leg will be reached but not for decreasing leg; for instance, see the case with 40 changes per day in Figure 8. This phenomena can also be explained by the timescales of each respective leg. Furthermore, as the number of flow changes per day approaches ∞, the WSE appears to approach the mean of the steady states. This convergence occurs faster for the more-downstream coordinates which can be seen in Figure 10. The cross-section where continuous dynamics is first observed for the five cases where it occurs have been plotted in Figure 11. For the 50 and 60 flow changes per day cases, this point is in close proximity to each other upstream of point 1. Since the width of the river in the lower parts would be reduced given the more frequent scenarios, it would likely affect the erosion and the morphology of the river.

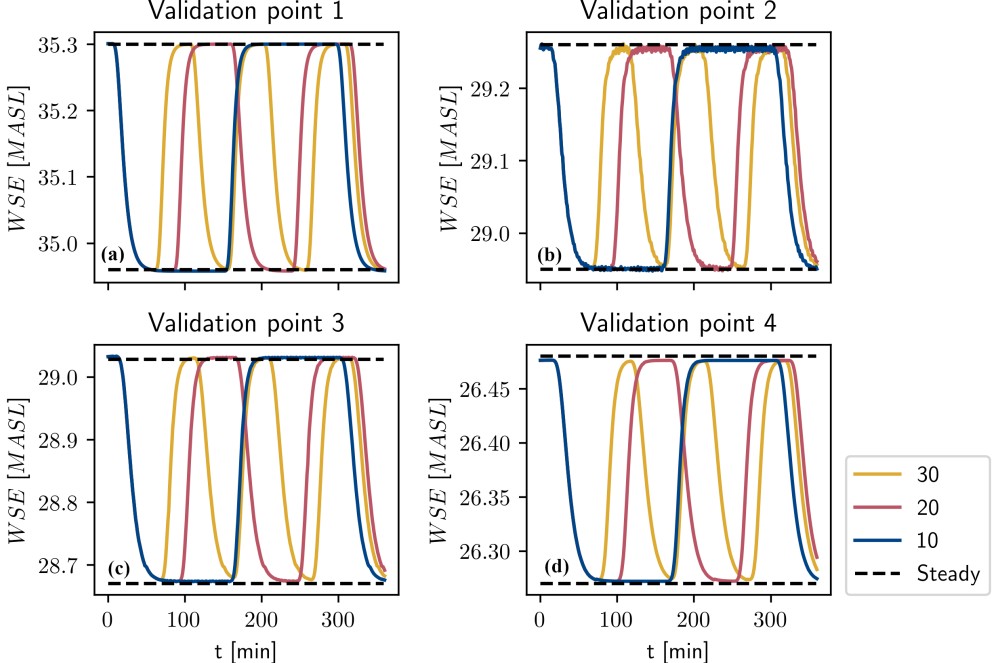

**Figure 7.** WSE for the points 1, 2, 3, and 4 given the flow scenarios with 10, 20, and 30 flow changes per day. (**a**) Validation point 1. (**b**) Validation point 2. (**c**) Validation point 3. (**d**) Validation point 4.

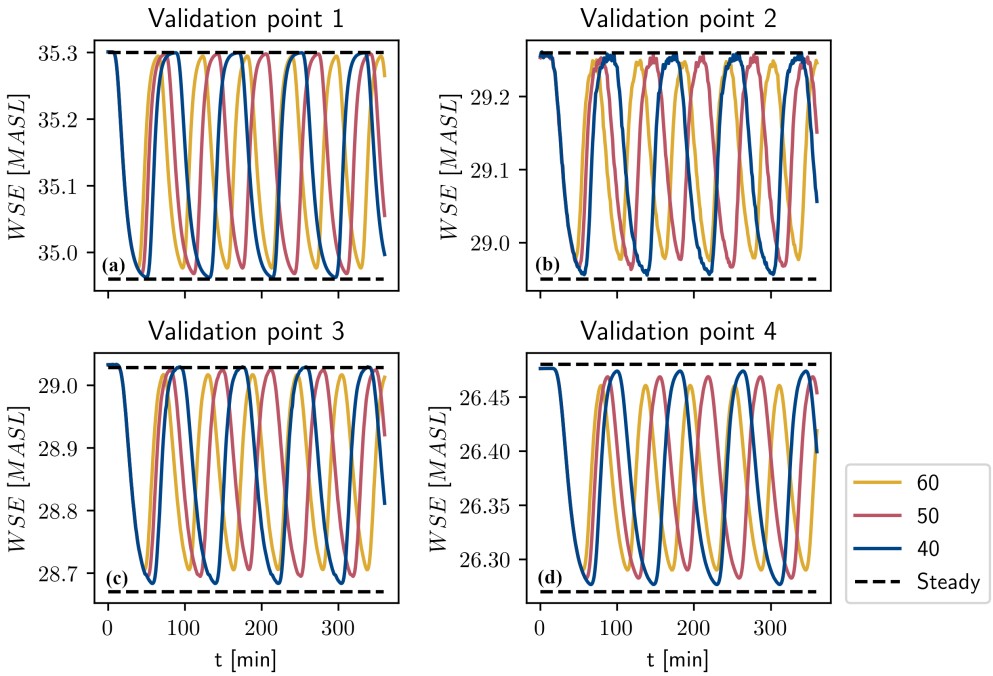

**Figure 8.** WSE for the points 1, 2, 3, and 4 given the flow scenarios with 40, 50, and 60 flow changes per day. (**a**) Validation point 1. (**b**) Validation point 2. (**c**) Validation point 3. (**d**) Validation point 4.

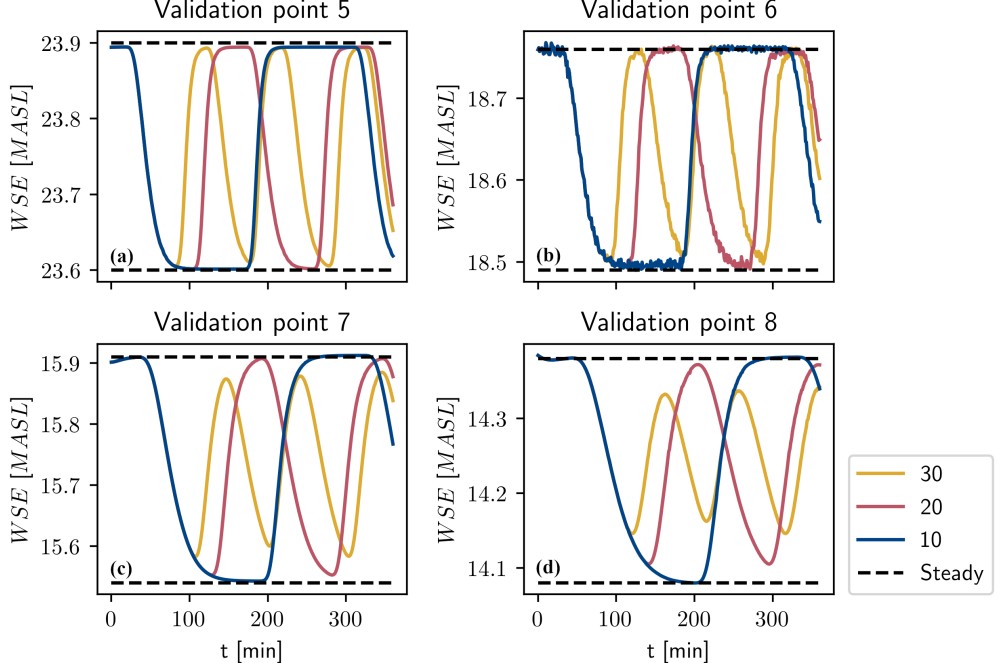

**Figure 9.** WSE for the points 5, 6, 7, and 8 given the flow scenarios with 10, 20, and 30 flow changes per day. (**a**) Validation point 1. (**b**) Validation point 2. (**c**) Validation point 3. (**d**) Validation point 4.

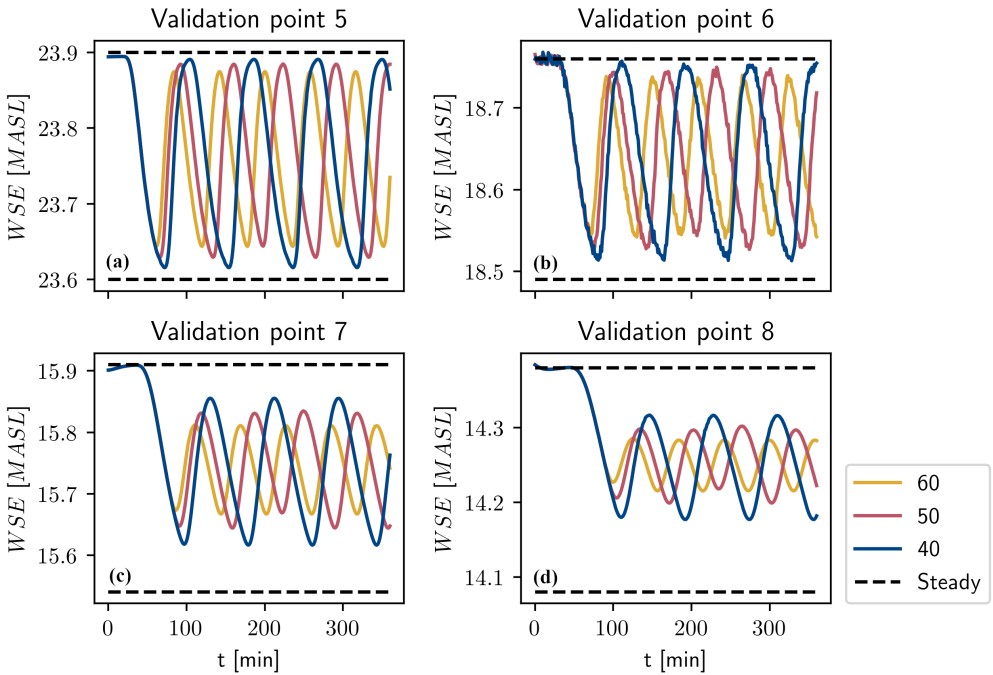

**Figure 10.** WSE for the points 5, 6, 7, and 8 given the flow scenarios with 40, 50, and 60 flow changes per day. (**a**) Validation point 5. (**b**) Validation point 6. (**c**) Validation point 7. (**d**) Validation point 8.

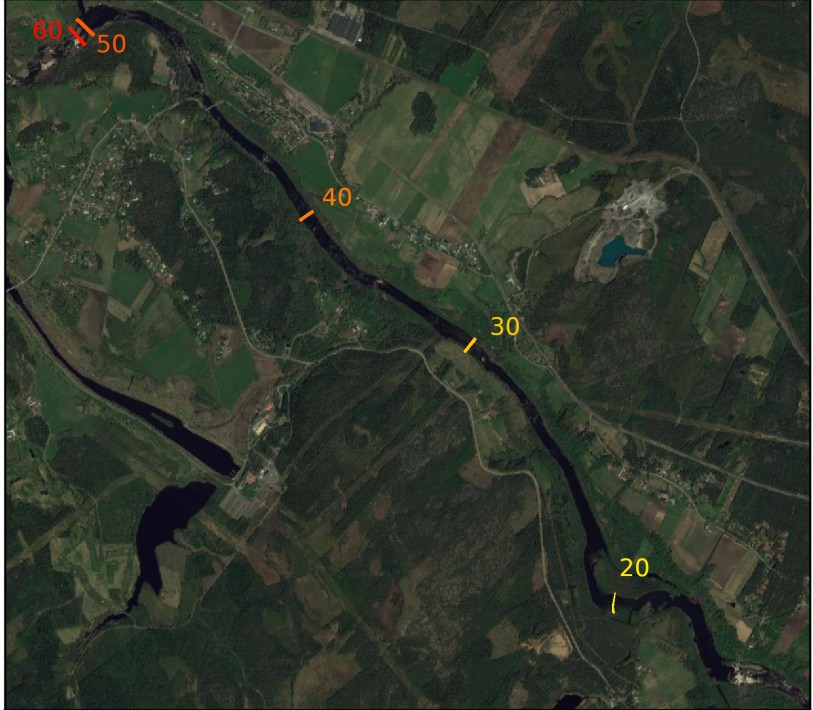

**Figure 11.** Cross-section corresponding to the most upstream observation of continuous dynamics for the five different scenarios where it occurs. The legend is in flow changes per day.

### 4.3. $A_{wetted}$ Dynamics for the Different Scenarios

The wetted area is computed each minute for all six scenarios using the methodology described in Section 3.6. The outcome has been plotted in Figure 12. The shape of all plots is directly related to the number of flow changes, see Figure 3. $A_{wetted}$ for 10 flow changes per day stands out from the rest, the maximum is significantly higher and the minimum is significantly lower than the rest of the plots. The remaining plots initially follow the same trajectory as the 10 flow changes per day cases,

$A_{wetted}$ diverges when the first change in discharge for each respective case happens. The 60, 50, 40, and 30 flow changes per day cases all increase for a short period of time before all begin to decrease. After this initial behavior, these cases begins to oscillate. This initial behavior is not observed for the 20 flow changes per day case, which appears to begin oscillating as soon as the first change in discharge occurs. Further, all cases except the 10 flow changes per day case appears to reach approximately the same maximum in the oscillating phase. This is not true for the minimum, except for the 30 and 40 flow changes per day cases, there is a clear difference where the 20 flow changes per day case has the lowest minimum and the 60 flow changes per day case has the highest minimum. Since the free-surface width in any cross-section in the reach is a function of the WSE, $A_{wetted}$ is also a function of the WSE. This reasoning in conjunction with Figures 7–10 gives us insights into the behavior of the local wetted area. For instance, the local wetted area in proximity to point 8 is likely not subject to dramatic fluctuations since the WSE oscillations are small. Conversely, in the vicinity around point 1, it is more likely to be large variations in local wetted area.

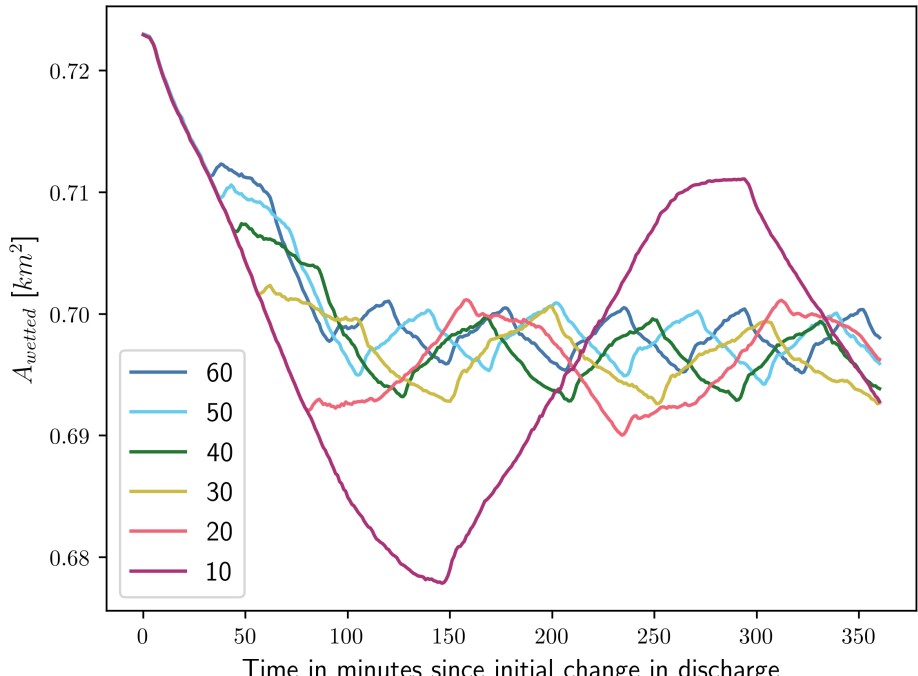

**Figure 12.** Total wetted area dynamics for the six different scenarios. The legend is in flow changes per day.

## 5. Conclusions

The aim of this study was to investigate the dynamics in the study reach when subject to different possible future hydropeaking scenarios. The SWE are able to accurately resolve the *increase* leg but not the *decrease* leg, as seen in Table 4. With this in mind, there are still valuable insights that can be gathered using the SWE to model transient river flows, taking the underestimated WSE damping into consideration. The WSE dynamics are dramatically affected by both the distance downstream of the spillways (Figure 6) as well as the frequency of the flow changes (Figures 7–10). With increasing frequency, the WSE and the wetted area far downstream are less affected by the rapid changes in discharge. If scenarios similar to the ones described in this work become more common in the future, then it is likely to be more detrimental to the habitats closer to the spillways rather than the habitats further downstream. It is possible that the habitats located further downstream will become more stable given more-frequent flow changes. Another consequence would be that the width of the river would be reduced, especially in the downstream parts, which could affect the erosion and morphology in the reach.

**Author Contributions:** The numerical simulations and analysis was performed by A.J.B. under supervision of A.G.A. and J.G.I.H. WSE measurements were performed by K.A. Manuscript was written by A.J.B. with assistance of A.G.A. and J.G.I.H. All authors have read and agreed to the published version of the manuscript.

**Funding:** This project has received funding from the European Union's Horizon 2020 research and innovation programme under grant agreement No 764011.

**Conflicts of Interest:** The authors declare no conflict of interest.

## Abbreviations

The following abbreviations are used in this manuscript:

| | |
|---|---|
| ACUR | Air Cushion Underground Reservoir |
| CFD | Computational Fluid Dynamics |
| DEM | Digital Elevation Model |
| MASL | Meters Above Sea Level |
| SWE | Shallow Water Equations |
| WSE | Water Surface Elevation |

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
