# Peer review of "Case Study of Transient Dynamics in a Bypass Reach"

_water, doi:10.3390/w12061585_

Round 1

Reviewer 1 Report

Hydraulic modeling is the future in both science and practical applications. The authors, using 3D modeling software, attempted to investigate various scenarios for changes in water surface elevations under the influence of a hydroelectric power plant.

The article presents high substantive quality, presents modern methods of forecasting issues important from the point of view of applicable regulations, including environmental protection and water management.

I would like to change the following elements that could increase the substantive value of the article:
1) On the first map, please mark the rivers, their flow direction, the scale indicating the map scale (or coordinates) and the north direction marked. In this shape, unfortunately, the map is unfortunately illegible and introduces some chaos in the perception of content. Additionally, on all maps, scale, north direction and legend should be added if necessary.
2) I would like to ask for information on where the location of the 8 selected points comes from? What were the criteria for their selection? I suggest comparing maps with Lidar, satellite images and the GPS points calibration performed by you (this is only a suggestion, although the selection of points in the analyzes should be justified).
3) In lines 87-93, please provide citation or citations in support of your justification for rejecting the inflow of the river on which the hydroelectric power plant is located.
4) Please add statistical analysis to the article - e.g. analysis of calculation uncertainty, but also basic statistical data for the calculated values ​​(in what ranges they fluctuated, what was the standard deviation, percentiles, median, etc.).
5) Please add DOI numbers in the bibliography to uniquely identify the cited literature sources.

In addition, I would like to ask you to answer the following issues (not necessary for improvement, but which may suggest further direction of research or adding some elements to the article):
1) Were the validation points selected with sufficient care? The analysis shows that the determined water levels were measured for about 250 minutes. Did you think to carry out tests in other conditions - also in medium and high waters? This would certainly allow for better calibration of the model and better selection of appropriate scenarios. Based on this type of statement, one could choose a representative wave containing the modal values ​​of the height of the water surface. Longer studies would certainly work in favor of the results obtained and reflect the values ​​obtained in the model compared to those in reality.
2) Can bathymetry be better reflected in the selection of the mesh? Are there other ways than what you presented?
3) Under different flow conditions, wetted area maps could be made.
4) What solutions could be used to eliminate errors in resolving the WSE decrease leg? This could be considered in the conclusions.
5) I lack information about hydrotechnical devices in the studied area. Information about the inflowing river appears unexpectedly. It is worth considering.

The article requires only minor corrections, after their introduction it will be suitable for publication in the journal Water. I wish the authors of the article all the best in their scientific pursuit.

Author Response

Thank you for your insight and your valuable comments, they have helped us to improve the manuscript significantly.

Point 1: On the first map, please mark the rivers, their flow direction, the scale indicating the map scale (or coordinates) and the north direction marked. In this shape, unfortunately, the map is unfortunately illegible and introduces some chaos in the perception of content. Additionally, on all maps, scale, north direction and legend should be added if necessary.

Response: The middle map of the lower river has now been removed. Furthermore a scale bar and a north arrow has been added to the map of the study reach. In this map a flow direction indicator has also been added.

Point 2: I would like to ask for information on where the location of the 8 selected points comes from? What were the criteria for their selection? I suggest comparing maps with Lidar, satellite images and the GPS points calibration performed by you (this is only a suggestion, although the selection of points in the analyzes should be justified).

Response: We have added a sentence justyfing the positioning of the divers in the reach in the "Bathymetry and depth measurements" section. A reference to a publication containing the GPS coordinates of the divers was added. If the reviewer believes that it would improve the article, these GPS coordinates could be added to this article also.

Point 3: In lines 87-93, please provide citation or citations in support of your justification for rejecting the inflow of the river on which the hydroelectric power plant is located.

Response: A sentence was rewritten to emphasize that the Vindel River merges upstream of the study reach, not in the study reach. The inflow of the Vindel River is hence not rejected.

Point 4: Please add statistical analysis to the article - e.g. analysis of calculation uncertainty, but also basic statistical data for the calculated values ​​(in what ranges they fluctuated, what was the standard deviation, percentiles, median, etc.).

Response: In section 3.4 Calibration, the statistical analysis that was tabulated inside Figure 4 has been moved to Table 1. The Pearson correlation is now refered to in the text. Standard deviation was also added to Table 1. Regarding the uncertainty of the model, this is discussed in section 3.7 Mesh study by using Richardson extrapolation on the wetted area.

Point 5: Please add DOI numbers in the bibliography to uniquely identify the cited literature sources.

Response: DOI numbers have been added to publications where they exist.

Additional Comments

Point 1: Were the validation points selected with sufficient care? The analysis shows that the determined water levels were measured for about 250 minutes. Did you think to carry out tests in other conditions - also in medium and high waters? This would certainly allow for better calibration of the model and better selection of appropriate scenarios. Based on this type of statement, one could choose a representative wave containing the modal values ​​of the height of the water surface. Longer studies would certainly work in favor of the results obtained and reflect the values ​​obtained in the model compared to those in reality.

Response:: The water levels were measured for a duration of approximately two months, the 250 minutes chosen represents a typical flow increase-decrease cycle. Unfortunately there was no other recorded water levels than for 50 and 21 m^3/s respectively. In the study there are usually no other flows except in the cases where spilling is needed i.e turbine downtime or during the spring flood. Indeed, more surface measurements would likely improve the calibration.

Point 2: Can bathymetry be better reflected in the selection of the mesh? Are there other ways than what you presented?

Response: Indeed the bathymery can be better represented by using a finer grid. The grid size used in this study is on the order of square meters but the bathymetry is on the order of square decimeters. Unfortunately it is not feasible to run the number of simulations run in this study with such a fine grid, in the future maybe it will be possible and hopefully it will give us a more complete picture of the flow.

Point 3: Under different flow conditions, wetted area maps could be made.

Response: Indeed, and this work is already under way and will hopefully be included in a not too distant future publication!

Point 4: What solutions could be used to eliminate errors in resolving the WSE decrease leg? This could be considered in the conclusions.

Response: This discrepancy was discussed in Section 4.1 and is believed to be due to the three-dimensional flow structures that is not resolved by the model. If the reviewer believe that inclusion of this would improve the conclusions, it can be added.

Point 5: I lack information about hydrotechnical devices in the studied area. Information about the inflowing river appears unexpectedly. It is worth considering.

Response: See Point 3 in the "Comments" section above regarding the Vindel River, the phrasing has been changed.

Reviewer 2 Report

I want to thank the authors for this work. It is interesting and well organized. The topic concern the dynamics in a bypass reach affected by hydropeaking. The research focus in a real case in Norway.

The topic of hydropeaking is one of the most important in modern literature due to the increase of hydropower production and other renewable energy and the interactin each other.

My comment regard the work are minor comments and some suggestions.

  1. In general the English is good but can be improved.
  2. in line 48, you can also talk about another aspect that it is not yet well studied. Depending on the river morphology, the hydropeaking can have also negative impact on human safety. for example Pisaturo, G.R., Righetti, M., Castellana, C., Larcher, M., Menapace, A., Premstaller, G.; A procedure for human safety assessment during hydropeaking events. (2019) Science of the Total Environment, 661, pp. 294-305. DOI: 10.1016/j.scitotenv.2019.01.158
  3. line 52. Another example for this aspect is the work.
    Premstaller, G., Cavedon, V., Pisaturo, G.R., Schweizer, S., Adami, V., Righetti, M.; Hydropeaking mitigation project on a multi-purpose hydro-scheme on Valsura River in South Tyrol/Italy. (2017) Science of the Total Environment, 574, pp. 642-653. DOI: 10.1016/j.scitotenv.2016.09.088
  4. line 64?. after "momentum equations" I suggest to add "for the three velocity components"
  5. in general I suggest in the figure to report the indication of the letters. for example in figure 1 you describe "a", "b" and "c" but you don't show which figure correspond.
  6. line 105. It is not clear from the figure 2, where are these abnormalities that you describe
  7. line 112. [32]. Hysteresis, ...
  8. line 119. "scenarios"
  9. figure 3. It is not clear how and why you choose this number of flow changes. For example e and f, 29 and 24 minute, seems too much in reality. I mean that I'm not sure that a hydropower plant can modify the flow rate so often.
  10. line 128. Even if the Manning number can be showed without units, I think that it is better to write the units of Manning number.
  11. Figure 6. increase arrow size if possible use the same color for the two graphs. y-y, r-r, b-b, k-k
  12. line 176. can you better explain? From figures 1 and 2 it is not clear this "poor geometry"
  13. line 185. Here, if you want, I suggest also to mention the fact that with a 3D model you are able to better simulate the vertical velocity profile and so the habitat analisys can have a improvement for example the results from
    Pisaturo, G.R., Righetti, M., Dumbser, M., Noack, M., Schneider, M., Cavedon, V.; The role of 3D-hydraulics in habitat modelling of hydropeaking events. (2017) Science of the Total Environment, 575, pp. 219-230. DOI: 10.1016/j.scitotenv.2016.10.046

Author Response

Thank you for your insight and your valuable comments, they have helped us to improve the manuscript significantly.

Point 1: In general the English is good but can be improved.

Response: The authors have made several improvents to grammar and sentence structure.

Point 2: in line 48, you can also talk about another aspect that it is not yet well studied. Depending on the river morphology, the hydropeaking can have also negative impact on human safety. for example Pisaturo, G.R., Righetti, M., Castellana, C., Larcher, M., Menapace, A., Premstaller, G.; A procedure for human safety assessment during hydropeaking events. (2019) Science of the Total Environment, 661, pp. 294-305. DOI: 10.1016/j.scitotenv.2019.01.158

Response: Thank you for this reference, it has been added to the introduction.

Point 3: line 52. Another example for this aspect is the work. Premstaller, G., Cavedon, V., Pisaturo, G.R., Schweizer, S., Adami, V., Righetti, M.; Hydropeaking mitigation project on a multi-purpose hydro-scheme on Valsura River in South Tyrol/Italy. (2017) Science of the Total Environment, 574, pp. 642-653. DOI: 10.1016/j.scitotenv.2016.09.088

Response: Thank you for this reference, it has also been added to the introduction.

Point 4: line 64?. after "momentum equations" I suggest to add "for the three velocity components"

Response: Equation 1 is written in vector formulation, hence all the velocity components are already implied by the "three momentum equations" statement. Furthermore, in the momentum equation we also solve for the pressure and the body forces, not uniquely the velocity components.

Point 5: in general I suggest in the figure to report the indication of the letters. for example in figure 1 you describe "a", "b" and "c" but you don't show which figure correspond.
Response: We agree that this can be confusing, however in the template instructions it is written "If there are multiple panels, they should be listed as: (a) Description of what is contained in the first panel. (b) Description of what is contained in the second panel." We have interpreted this in a way so that the leftmost panel corresponds to (a). Hopefully this explanation is sufficient.

Point 6: line 105. It is not clear from the figure 2, where are these abnormalities that you describe

Response: We have clarified this part of the manuscript, hopefully it is more clear now.

Point 7: line 112. [32]. Hysteresis, ...

Response: We have change the language according to your comment.

Point 8: line 119. "scenarios"

Response: Corrected according to the reviewers suggestion.

Point 9:figure 3. It is not clear how and why you choose this number of flow changes. For example e and f, 29 and 24 minute, seems too much in reality. I mean that I'm not sure that a hydropower plant can modify the flow rate so often.
Response: You are correct, the reason why there are so many changes is because it is an artifical goal of the HydroFlex project to investigate up to 30 start / stops per day. We realize that we haven't sufficiently explained this. An explanation of the HydroFlex consortium has been added to the introduction.

Point 10: line 128. Even if the Manning number can be showed without units, I think that it is better to write the units of Manning number.

Response: We agree and units have now been added.

Point 11: Figure 6. increase arrow size if possible use the same color for the two graphs. y-y, r-r, b-b, k-k

Response: Both graphs have been corrected.

Point 12: line 176. can you better explain? From figures 1 and 2 it is not clear this "poor geometry"

Response: Hopefully the changes made for Point 6 will better explain this area of poor geometry.

Point 13: line 185. Here, if you want, I suggest also to mention the fact that with a 3D model you are able to better simulate the vertical velocity profile and so the habitat analisys can have a improvement for example the results from Pisaturo, G.R., Righetti, M., Dumbser, M., Noack, M., Schneider, M., Cavedon, V.; The role of 3D-hydraulics in habitat modelling of hydropeaking events. (2017) Science of the Total Environment, 575, pp. 219-230. DOI: 10.1016/j.scitotenv.2016.10.046

Response: Added a reference to this work in section 4.1.

Round 2

Reviewer 1 Report

All comments contained in the review were taken into account by the authors of the article, and in the event of inability to perform - properly argued.

In the summary authors could only add information on solutions that could be used to eliminate errors in resolving the WSE decrease leg.

In my opinion, however, the article in this form is already suitable for publication and falls within the scope of the journal Water.

I wish the authors good luck in their further scientific and research work.

Reviewer 2 Report

I want to thanks the authors for their work and for the corrections that they have made.